# An Efficient and Credible Multi-Source Trust Fusion Mechanism Based on Time Decay for Edge Computing

**Wenping Kong [1] , Xiaoyong Li [1,2,*], Liyang Hou [1] and Yanrong Li [1]**

1   School of Cyberspace Security, Beijing University of Posts and Telecommunications, Beijing 100876, China;
kwenping@163.com (W.K.); leonhoou@163.com (L.H.); wangwang5780@outlook.com (Y.L.)
2   Key Laboratory of Trustworthy Distributed Computing and Service, Ministry of Education,
Beijing University of Posts and Telecommunications, Beijing 100876, China
*   Correspondence: lixiaoyong@bupt.edu.cn; Tel.: +86-1590-102-8066

**Abstract:** With the development of 5G, user terminal computing moves up and cloud computing sinks, thus forming a computing fusion at the edge. Edge computing with high-efficiency, real-time, and fast features will become part of 5G construction. Utilizing distributed computing and storage resources at the edge of the network to perform distributed data processing tasks can alleviate the load on the cloud computing center, which is also the development trend of edge computing. When a malicious node exists, the error information feedback by the node will affect the result of local perception decision. To solve the problem of malicious behavior of the node, a node trust evaluation mechanism of interactive behavior is introduced. The trust mechanism for edge computing network environments is introduced as a novel security solution. First, the key thought of the trust mechanism proposed in this paper is to establish a trust relationship between edge nodes in open edge computing environment. Then, a multi-source trust fusion algorithm based on time decay aggregates direct interaction trust and different third-party recommendation trust to calculate the global trust of the evaluated nodes. Finally, simulation experiments show that the algorithm has a certain degree of improvement in computational efficiency and interaction success rate over other existing models, which reduces the situation of malicious node deception.

**Keywords:** edge computing; 5G; multi-source trust fusion mechanism; trust computing

## 1. Introduction

With the rapid development of 5G network architecture, new service models and businesses such as intelligent transportation, smart city, location services, and mobile payments are constantly emerging. The number of devices accessing the network has increased dramatically, and the data transmitted in the network are also growing geometrically [1–3]. In the three major scenarios of 5G, low-latency Internet vehicles, automatic driving, and public security cameras throughout the smart city all need edge computing because of the processing demand of massive data, including some typical scenarios such as intelligent manufacturing and intelligent medical treatment. With the advent of the 5G era, the intelligence connection of things really can come true. However, it puts forward new requirements for computing structure, which requires low latency, large bandwidth, high concurrency, and localization. Traditional cloud computing centers have been unable to meet the low-latency, intensive network access, and service requirements [4]. Using distributed computing and storage resources on the edge of the network to perform data distributed task processing and alleviate the load of cloud computing center will become the key to the development of 5G [5,6], as shown in Figure 1.

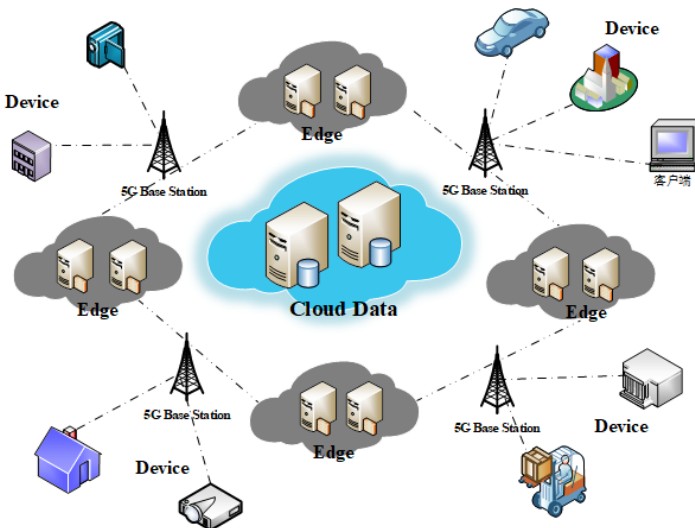

**Figure 1.** 5G edge computing scenario.

Edge computing is an open platform that integrates core capabilities such as networking, computing, storage, and applications. It provides high-bandwidth, low-latency edge intelligent, services nearby at the edge of the network near the source of people, things, or data. At the same time, it forms a localized deployment through service sinking, which effectively reduces the requirements of network return bandwidth and network load.

### 1.1. Motivation

Edge computing technology combines IT service environment and cloud computing technology at the edge of the network to improve the computing and storage capabilities of edge networks, reduce network operation and service delivery delay, and improve user service quality experience. For this type of application [7,8], the rise of edge computing has also attracted a lot of attention to the security of users, data, and computing nodes in the network. Although edge computing pushes calculations closer to users, it avoids data being uploaded to the cloud, reducing the possibility of private data leakage. However, compared to cloud computing centers, edge computing devices are usually close to the user side or transmission path, which has a higher potential to be attacked. Therefore, the security of the edge node itself is still a problem that cannot be ignored. Due to the distributed and heterogeneous characteristics of edge computing nodes, it is difficult to manage them uniformly, which makes resource nodes vulnerable to malicious attacks [9,10].

Generally speaking, the purpose of network security is to provide a trust protection mechanism [11], which can prevent the protected subject from being attacked and illegally accessed by malicious subjects. The introduction of trust mechanism can better ensure the security and collaboration of Internet application systems [12,13]. Data storage and computing tasks in edge computing mostly depend on edge nodes, many of which are exposed to the natural environment. However, unlike cloud data centers, edge computing does not have stable infrastructure protection. Therefore, the realization of a secure and trusted resource sharing computing environment is one of the key elements of the development of edge computing.

### 1.2. Contribution

To solve the malicious behavior of nodes, the trust evaluation mechanism of nodes with interactive behavior is introduced [14]. The trust mechanism based on node interaction behavior provides dynamic behavior awareness during the service provision process, which can effectively prevent malicious service behavior from the authentication service provider [15]. The main content of the trust mechanism

includes collecting node's credibility, calculating node's trust according to collecting evaluation records, and deciding whether to interact based on the trust value of the node.

Combined with edge computing in 5G environment, trust between nodes may change dynamically over time, and it has the characteristics of time lag. In this paper, a multi-source trust fusion algorithm is designed by aggregating direct trust based on time decay, recommendation trust that interacts with requesting nodes, and recommendation trust based on *Jaccard* similarity. Simulation experiments show that the multi-source trust fusion algorithm proposed in this paper has better computing efficiency, and the introduction of trust mechanism improves the ability to detect malicious nodes. The multi-source trust fusion scheme is superior to the existing methods in the following three aspects.

1.  The multi-source trust mechanism frame in edge computing: We assert that the global trust of nodes consists of two parts: direct trust and two kinds of recommendation trust. Direct trust is that the request node and the service node directly interact based on the time attenuation. There  are two types of recommendation trusts: one is the direct recommendation trust with interaction between the recommendation node and the request node, and the other is the recommendation trust based on the similarity measure of the same device when there is no interaction between the recommendation node and the request node. Recommendation trust is an effective way to establish a reputation-based trust relationship between edge nodes. It integrates direct trust and a variety of recommendation trust to form the global trust of nodes.
2.  Trust mechanism based on time decay: Time-based trust refers to the trustworthiness degree of the trust evaluator in the open free interactive network environment, according to his own direct interaction experience or other recommender's recommendation information in the effective time domain, whether the trusted person can complete a specific service or transaction capability honestly, safely, and reliably as expected.  Attributes that decay over time are introduced, improving the accuracy of trust quantification.
3.  An effective and secure algorithm of trust update and reward and punishment: The edge computing environment needs to update the trust of the interaction nodes after each interaction to reflect the trust changes of the interaction nodes more objectively.  Each time the trust is updated, the nodes should be rewarded and punished for trust according to the performance of the node's behavior in this interaction. The nodes that perform well should be appropriately rewarded after the interaction, and the nodes that perform poorly should be punished after the interaction.The reward and punishment mechanism can prevent excessive rewards due to increased node satisfaction when swaying attacks occur, stimulate active and benign nodes to participate in interaction, and build a more secure and healthy network environment.

The key idea of the trust mechanism proposed in this paper is to establish trust relationship between edge nodes in the open edge computing environment by fusing three different trust methods. At the same time, we adopt trust evaluation mechanism to evaluate the collaborative efficiency of nodes, which are suitable for large-scale edge computing and are helpful for communication and cooperation between nodes in the edge layer.

The rest of this paper is organized as follows: In Section 2, we introduce the related work. Section 3 describes the model framework and trust relationship with edge computing. Section 4 outlines the multi-source trust fusion algorithm. Section 5 presents the experiment. Section 6 summarizes the full text and puts forward the improvement methods.

## 2. Relevant Work

The core concept of edge computing is a distributed computing form that places computing resources and data storage at the edge nodes of the network [16] and supports new human-centric services by providing users with low-latency service awareness and controllable network transmission costs [17]. It gives full play to the advantages of large broadband, low latency, and wide link in 5G network [18]. Facing heterogeneous resources and system sharing in edge computing environment,

we need to consider such factors [19] as matching of resource performance, behavior trust, identity trust and resource providing ability trust to effectively face the challenge of interaction and sharing between ubiquitous terminal nodes in the network.

As we can see, trust plays an important role in many applications and people's life [20]. We can evaluate the trustworthiness of the nodes in the network by establishing an effective trust mechanism to solve the problem of network security. At present, there are few studies in the field of edge computing, and most of them only focus on the field of mobile cloud computing. Although trust computing is relatively new in edge computing, much research work has been developed and explored in edge computing. This section discusses the research progress in this field.

Zhang et al. [21] discussed and analyzed the architecture of fog computing and pointed out related potential security and trust issues. They comprehensively reported on how to solve these problems in the existing literature and discussed open challenges in security and trustworthiness of fog computing, research trends and future topics.

The combination of blockchain and SDN [22] is analyzed to ensure the effective operation of the VANET system in the 5G and fog computing paradigms. Sharing management responsibilities between the blockchain and SDN helps reduce the pressure on the controller due to the ubiquitous processing that occurs. A trust-based model is also proposed to suppress malicious activities in the network.

Hussain et al. [23] proposed a trust management scheme, studied how to calculate the reputation of edge data center, and allowed mobile users to use cloud computing through mobile cloud providers. With centralized trust management, users of this system can evaluate cloudlet services anonymously.

Kantert et al. [24] proposed a self-maintenance trust management system, which enables autonomous servers from different administrative domains to share their resources in grid-like situations. Contrary to other grid deployments, it assumes that there are selfish individuals or malicious service providers. Therefore, Kantert et al. designed a set of trust metrics computed in an autonomous and distributed way to evaluate and manage trust in computing edge data centers.

Huang et al. [25] introduced a vehicle networking generated by improving the computing power of vehicle network edges, and proposed DREAMS for safe and efficient vehicles. In DREAMS, responsibility for reputation management shifts to the vicinity of mobile vehicles to improve overall performance. The geographically distributed LAS are responsible for local reputation management, and all valid reputation parts are collected into LAS to ensure that they are responsible for reputation renewal. According to familiarity, similarity, and timeliness, different reputation subdivisions are weighted to obtain higher accuracy. Compared with the traditional reputation computing model, DREAMS has great advantages in optimizing bad behavior detection and improving the recognition rate of bad vehicles.

Ruan et al. [26] proposed a flexible security architecture of Internet of Things access to edge cloud computing. Through multi-access edge cloud computing platform, the Internet of Things can connect to a nearby cloudlet, and then it can communicate with other devices, computers, or cloud resources. They proposed and applied a trust management framework based on measurement theory, which can evaluate the trust in applications and computing resources. In addition, the framework can help vendors dynamically configure resources based on real-time trust information.

Yuan et al. [12] proposed a reliable and lightweight trust mechanism for IoT edge devices based on multi-source feedback information fusion. It mainly constructs the trust model for the Internet of Things environment. The agent aggregates the direct trust between device $a$ and device $b$ and the feedback trust between the remaining devices and the evaluated device $b$, forming the global trust of the evaluated device $b$. It uses a feedback information fusion algorithm based on objective information entropy theory, which is manually weighted. This trust mechanism mainly uses task failure rate method to evaluate security and reliability. However, the indirect trust in this paper comes from two kinds of recommendation trust for node $j$; one is the direct recommendation trust that the recommendation entity $k$ interacts with both the request node $i$ and the evaluated node $j$, and the other

is the recommendation trust that interacts with the evaluated node *j* but does not interact with the request node *i* based on the similarity of the same device interaction.

As mentioned above, although the feedback mechanism is undoubtedly a basic requirement of the trust system, the recent research ignores the collusion problem caused by the feedback mechanism itself, which will greatly reduce the reliability of the trust system.

In addition, the previous studies mostly used subjective methods to assign weights to trust decision-making factors, which cannot reflect the adaptability of trust decision-making process, and may lead to misjudgment of trust calculation. This paper proposes a multi-source trust fusion mechanism based on time decay for edge computing, which guarantees the security of edge computing environment.

## 3. Edge Computing Architecture with Trust Mechanism

In this section, we first introduce a trust model framework that multi-source trust fusion mechanism based on time attenuation for edge computing. The trust relationship of nodes in edge computing is also discussed.

### 3.1. Edge Computing Architecture

Edge computing [27] refers to a new computing model that performs calculations at the edge of the network. In edge computing, the downlink data of the edge represents the cloud service, and the uplink data represent the Internet of Everything service. The edge of edge computing refers to any computing and network resources from the data source to the cloud computing center path. By offloading some of the computing tasks from the cloud data center to the edge server while processing the data at the edge, the application can reduce latency and respond more quickly to user service requests. Edge calculation has obvious advantages as follows;

1.  processing a large amount of temporary data at the edge of the network and no longer uploading to the cloud, which greatly reduces the pressure on network bandwidth and data center overhead; and
2.  data processing is performed near the data producer, and there is no need to request the response of the cloud computing center through the network, which greatly reduces the system delay and enhances the service response capability.

As shown in Figure 2, the edge computing architecture consists of three layers [28–30]: the cloud edge collaborative joint network architecture can generally be divided into terminal device layer, edge computing layer, and cloud computing layer. Each layer can communicate between layers and across layers. The composition of each layer determines the computing and storage capacity of each layer, and thus determines the functions of each layer [31].

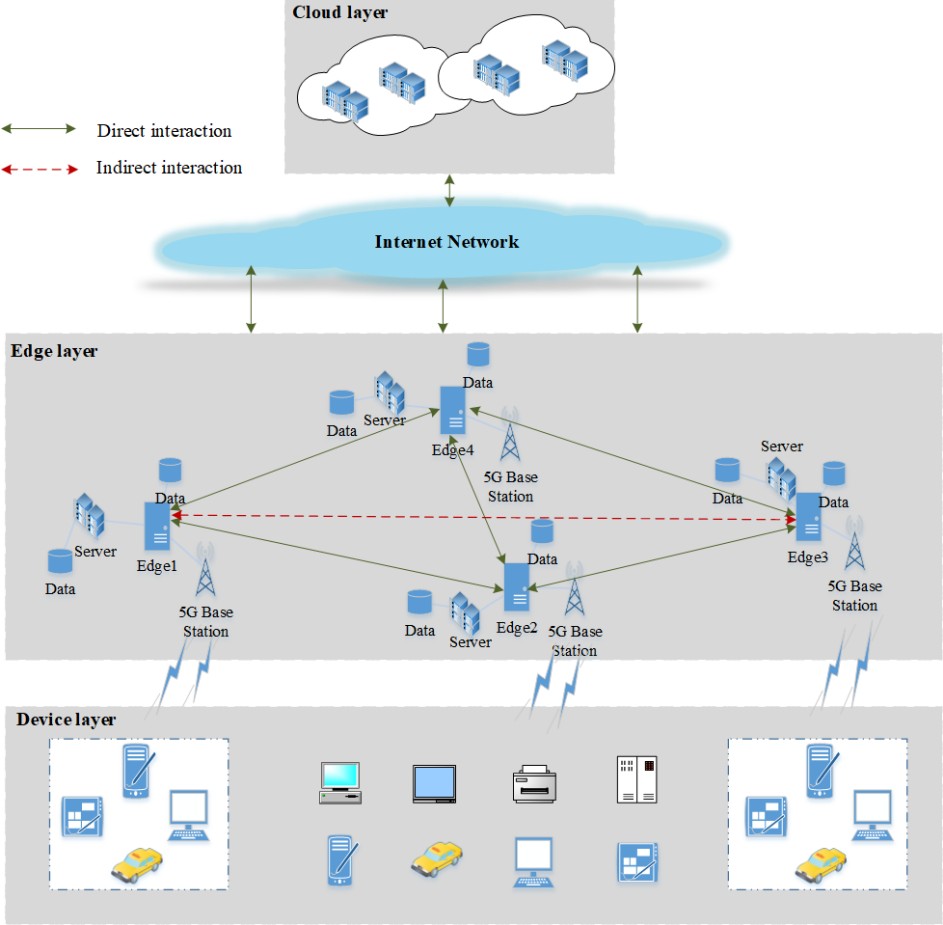

**Figure 2.** Edge computing architecture.

### 3.1.1. Terminal Device Layer

The terminal device layer is composed of various Internet of Things devices (such as sensors, cameras, smart phones, etc.), which mainly collect raw data and report them [32]. In the terminal device layer, only the sensing capabilities of various IoT devices are considered, regardless of their computing power. Billions of IoT devices at the terminal layer continuously collect various types of data, and use the form of event sources as input to application services. After the terminal device layer and the edge node complete the interactive behavior, the mutual evaluation information is submitted.

### 3.1.2. Edge Computing Layer

The edge computing layer is composed of network edge nodes, which are widely distributed between the terminal equipment and cloud computing center. It can be the intelligent terminal equipment itself, such as smart bracelet, smart camera, etc., or it can be deployed in the network connection, such as gateway, router, and so on. Obviously, the computing and storage resources of the edge nodes are very different, and the resources of the edge nodes are dynamic. For example, the available resources of the intelligent bracelet are dynamic with the use of people. The edge computing layer realizes the basic service response through the reasonable deployment and deployment of the network edge computing and storage capacity. After the interaction between the edge nodes, the mutual evaluation information is submitted.

### 3.1.3. Cloud Computing Layer

Cloud computing is still the most powerful data processing center in the combined services of cloud computing. The data reported by the edge computing layer will be stored permanently in

the cloud computing center. Analytical tasks that cannot be processed by the edge computing layer and processing tasks for comprehensive global information still need to be completed in the cloud computing center. In addition, the cloud computing center can dynamically adjust the deployment strategy and algorithm of edge computing layer according to the distribution of network resources.

*3.2. Analysis of Trust Relations in Edge Computing*

In the proposed trust mechanism scheme for edge computing, trust computing is completely completed by the edge layer, which reduces the burden on the cloud data center and makes it more efficient to perform in the network. Otherwise, all feedback must be sent to the cloud data center for processing, and the response time is too long, which increases the pressure on the network. In the process of performing trusted computing at the edge, a trusted execution environment [33] needs to be established to ensure the security of the edge layer services to reduce latency, improve execution efficiency, and reduce network pressure.

According to the role of network nodes in edge computing, two sources are provided, including edge node sets and device sets. Therefore, two sets of entities can be formed: an edge node set (E), where i is the ID of the edge node and n is the total number of edge nodes, and a device set (D), where p is the device ID and m is the total number of devices. The trust in the trust mechanism comes from the service request node in the interaction record to predict the trustworthiness of the service node, and the prediction comes from the subjective judgment of the service request node, depending on whether the effect reaches the subjective expectations of the service request node, such as the service quality rating degree in the interaction process. Trust is based on the historical interaction records of nodes, and the interaction between different nodes in the network is dynamically updated in real time. Trust is based on the historical interaction records of nodes, and the interaction between different nodes in the network is dynamically updated in real time. In each interaction, the degree of service satisfaction is different, and the degree of trust for different nodes is also different. In the model, the important factors that affect trust can be abstracted, and the degree of trust is usually normalized.

**Definition 1.** *Trust is the subjective judgment or expectation of an entity according to the existing knowledge to the satisfaction degree of another entity's interaction behavior or service. In this model, trust evaluation values are normalized to intervals [0,1]. Trust can be defined as a binary mapping on the universe:*

$$E \times E \rightarrow [0, 1] \tag{1}$$

When a malicious entity with a low trust evaluation usually wants to obtain high trust through an interactive performance in order to deceive trust, the value of the evaluation information is normalized, so that the malicious entity's plot cannot be succeeded, and the false evaluation of too high or too low will affect the accuracy of trust.

Trust changes over time. As time goes by, the reference value of trust information to current trust evaluation is getting weaker, showing a monotonous decreasing trend. The longer is the trust evaluation, the worse is the persuasiveness. To reflect the timeliness and dynamics of trust evaluation, a dynamic time window is introduced to describe the behavior characteristics of entities in a period of time, and only the interactive behavior trust records in the time window are considered in trust evaluation.

**Definition 2.** *(Time attenuation) The time attenuation refers to the attenuation degree of the influence of the entity trust information on the current entity trust evaluation over time.*

Introducing a time attenuation factor [34] can effectively protect the importance of recent interactions. The time can be expressed by a class of monotone decreasing functions. The longer it is from the current time, the less impact the interaction results have on the current. The time attenuation

is expressed by the attenuation function $\varphi(x)$, and the attenuation function should have the following properties.

(1)  $\forall x \in (0, \infty), \varphi(x) \in (0, 1]$;
(2)  if $x_1 < x_2$, then $\varphi(x_1) > \varphi(x_2)$; and
(3)  $\varphi(0) = 1$.

The time interval from the current time $t_r$ forward to a certain historical interaction time $t$ is denoted as $\Delta t = t_r - t$, and the time attenuation is represented by the attenuation function as follows:

$$\varphi(\Delta t) = e^{-\frac{\Delta t}{\lambda}} \tag{2}$$

The parameter $\lambda$ adjusts the decay rate of historical trust over time, which should be set according to the specific application. Generally, $\lambda = \frac{T_0}{ln2}$ is set, so that the half-life trust is attenuated by half, and $T_0$ is the trust half-life. If 60 time slices are the half-life of trust decay over time, then choose $\lambda = \frac{60}{ln2}$.

Trust in edge computing can be divided into direct trust and recommendation trust according to their sources. The trust relationship is shown in Figure 3.

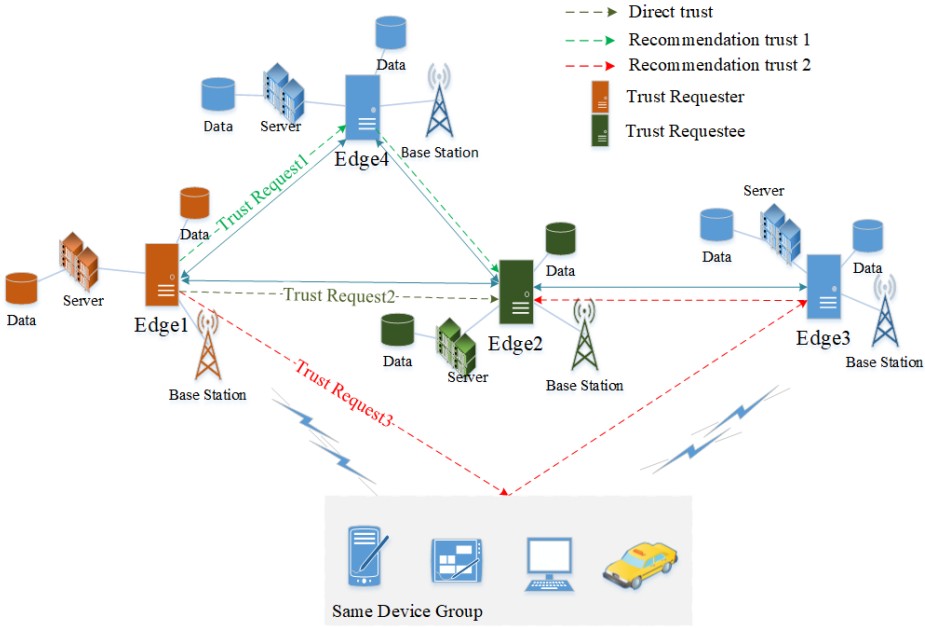

**Figure 3.** Trust relationship in edge computing.

**Definition 3.** *(Direct Trust) $DT_{ij}$ is the trust evaluation of $e_j$ based on the direct interaction between node $e_i$ and node $e_j$. It is based on the historical record of interaction between nodes.*

When node $e_i$ evaluates node $e_j$, the interaction history evaluation of itself and node $e_j$ may not be objective enough. It also needs recommendation trust information of other nodes. In particular, $e_i$ and $e_j$ have never had interactions. When node $e_i$ accepts the recommendation, it is also necessary to consider whether the recommended trust is trustworthy.

**Definition 4.** *(Recommendation Trust) Node $e_i$ determines the trust of the entity formed by the third party entity k providing a trust evaluation of node $e_j$.*

**Definition 5.** *(Recommendation credibility) Recommendation credibility is the credibility of information recommended by entity k as a recommender by node $e_i$, expressed in $Cr_{ik}$.*

Node $e_i$'s understanding of recommendation entity $k$ comes from two situations.

1.  There has been direct interaction between node $e_i$ and recommendation entity $k$, and the recommendation credibility of recommendation entity $k$ can be inferred from the satisfaction of direct interaction.
2.  There has been no interaction between node $e_i$ and recommendation entity $k$. Both $e_i$ and recommendation entity $k$ have evaluated the trust of several common objects. $e_i$ can estimate recommendation credibility of recommendation entity $k$ based on the similarity of evaluation.

**Definition 6.** *(Global Trust) $e_i$ can be obtained from the global trust value of $e_j$ by weighted averaging according to the direct trust obtained from the direct interaction with $e_j$ and the two kinds of recommendation trust about $e_j$ obtained from other recommendation entities.*

In the traditional trust computing system, trust mainly comes from the evaluation of direct interaction, which may bring many problems, and is not objective enough. It may also lead to the aggressiveness of trust and the sparseness of evaluation. Relevant trust models mostly focus on a single type of recommendation, and do not integrate several main types of trust recommendation, which can not effectively solve the problem of diversity of trust recommendation. In this scheme, the addition of time decay and multiple sources of trust can effectively identify and resist fraud and attack behaviors from various forms.

*3.3. Algorithm Description*

According to the analysis of the trust source, the trust mechanism proposed in this paper needs to calculate the global trust degree $G_{ij}$ of the evaluated node $e_j$. $DT_{ij}$ means direct trust between $e_i$ and $e_j$. $RT_{ij}^{(1)}$ indicates that $e_i$ interacts with the third-party recommendation entity $k$ for the recommendation trust of node $e_j$. $RT_{ij}^{(2)}$ means that $e_i$ has no interaction with the third-party recommendation entity $k$ for the recommendation trust of node $e_j$.

The steps of the global trust algorithm of the evaluated nodes in the edge calculation are as follows.

**Input:** A node set $(E = e_1, e_2, \cdots, e_n)$, device set $(D = d_1, d_2, \cdots, d_m)$, and time attenuation factor $\varphi(\Delta t)$.

**Output:** $G_{ij}$.

Step 1: Assign an initial trust degree of $T_0 = 0.5$ to each new node joining the trust network, and the initial trust is updated after a transaction with the node.

Step 2: Node $e_i$ calculates the direct trust degree $DT_{ij}$ of node $e_j$ according to the satisfaction evaluation performed by itself when interacting with node $e_j$.

Step 3: Node $e_i$ sends a request for recommendation trust information about node $e_j$ from other recommendation entity $k$. According to the interaction between recommendation entity $k$ and $e_i$, the recommendation trust $RT_{ij}^{(1)}$ for $e_j$ is calculated; recommendation entity $k$ has no interaction with node $e_i$, and the recommendation trust $RT_{ij}^{(2)}$ for $e_j$ is calculated according to the similarity of the rating of the same devices, and two kinds of recommendation trust $RT_{ij}$ are calculated by aggregation.

Step 4: Aggregating direct trust $DT_{ij}$ and recommendation trust $RT_{ij}$, and calculate the global trust $G_{ij}$ of node $e_i$ to node $e_j$.

Step 5: Implement dynamic trust update of node $e_i$ to node $e_j$.

## 4. Trust Calculation

According to the above definition, in the proposed trust computing mechanism, there are direct trust relationship and two different recommended trust relationships, and different trust source calculation methods are different. In this section, we introduce the relevant computing mechanism for these trust factors.

### 4.1. Direct Trust

Direct trust $DT_{ij}$ is the trust evaluation of $e_j$ based on the direct interaction between $e_i$ and $e_j$.

Suppose $e_i$ and $e_j$ interact n times in the time window. After the *l*th interaction, $e_i$ evaluates $e_j$ to get the evaluation value $S_l \in [0, 1], l = 1, \cdots, n$, and then the direct trust is calculated as follows.

$$DT_{ij} = \frac{\sum\limits_{l=1}^{n} S_l \varphi(\Delta_{t_l})}{\sum\limits_{l=1}^{n} \varphi(\Delta t_l)} \tag{3}$$

$\varphi(\triangle t_l)$ is the trust attenuation of the *l*th interaction relative to the current moment. It reflects the attribute that the trust relationship attenuation with time.

### 4.2. Recommendation Trust

When node $e_i$ evaluates node $e_j$, the history of interaction between itself and node $e_j$ may not be objective enough. It also needs other nodes to recommend the trust information of $e_j$, especially when $e_i$ and $e_j$ have never interacted.

#### 4.2.1. Recommendation Trust Based on Direct Interaction

Node $e_i$ has interacted with recommendation entity $k$, and the set of recommendation entities that have interacted is recorded as $RS_1$. Any recommendation entity $k \in RS_1$ feeds its trust degree to $e_i$. Node $e_i$ uses the direct trust $DT_{ik}$ to the recommended entity $k$ as the recommendation credibility of the entity $k$, that is,

$$Cr_{ik}^{(1)} = DT_{ik} \tag{4}$$

The recommendation trust generated by this type of recommendation is as follows:

$$RT_{ij}^{(1)} = \frac{\sum\limits_{k \in RS_1} Cr_{ik}^{(1)} T_{kj} \varphi(\triangle t_k)}{\sum\limits_{k \in RS_1} Cr_{ik}^{(1)} \varphi(\triangle t_k)} \tag{5}$$

Where $DT_{ik}$ represents the direct trust of $e_i$ to the recommended entity $k$, and $T_{kj}$ represents the trust of the recommended entity $k$ to the evaluated node $e_j$. $\varphi(\Delta t_k)$ is the trust attenuation of the historical interaction between $e_i$ and the recommended entity $k$.

#### 4.2.2. Recommendation Trust Based on Evaluation Similarity

There is no direct interaction between node $e_i$ and the recommendation entity $k$, but they have evaluated the trust of many same objects. The set of recommendation entity $k$ is recorded as $RS_2$, and node $e_i$ can estimate the feedback credibility of the feedback trust of the recommended entity based on the evaluated vector similarity.

Two entities evaluate the trust of the same group of entities, and the evaluation results are two sets of vectors. The matching measure of these two sets of vectors reflects the similarity of the two groups of trust evaluation. Computational similarity between nodes can be better used to prevent malicious nodes from colluding feedback scoring, reputation smashing, etc., and it is easy to detect dishonest feedback from real feedback.

**Definition 7.** *(Vector similarity) Vector similarity denotes the matching degree of two sets of vectors. Vector similarity can be used to measure the behavior similarity of two entities.*

**Definition 8.** *(Jaccard similarity coefficient) Jaccard similarity coefficient is an index used to measure the similarity of two sets, which is defined as the number of elements in the intersection of two sets divided by the number of elements in the union. Jaccard similarity measures similarity by comparing the proportion of common characteristics between the two.*

Feedback credibility is calculated using *Jaccard* similarity. *IS* represents the set of entities that have interacted with $e_i$, *KS* represents the set of entities that interacted with the recommended entity $k \in RS_2$, and $CS = IS \cap KS$ is the set of entities that have interacted with both $e_i$ and recommended entity $k$. The calculated *Jaccard* similarity is used as the feedback credibility of the recommended entity $k$ by comparing node $e_i$ to the trust score value of each entity in the set CS and the trust score value of the recommended entity $k$ to each entity in the set $CS$.

The trust evaluation vectors of the nodes $e_i$ and the recommended entity $k$ for the entities in the set $CS$ are, respectively, recorded as $(T_{i1}, T_{i2}, \cdots, T_{it})$ and $(T_{k1}, T_{k2}, \cdots, T_{kt})$, and the feedback credibility of the recommended entity $Cr_{ik}^{(2)}$ can be measured by the following vector similarity,

$$Cr_{ik}^{(2)} = \eta \frac{2 \sum\limits_{p \in CS} T_{ip} T_{kp}}{\sum\limits_{p \in CS} T_{ip}^2 + \sum\limits_{p \in CS} T_{kp}^2} + (1 - \eta) \frac{|CS|}{|IS \cup KS|} \tag{6}$$

where $\eta \in (0.5, 1]$ control vector similarity calculation obtains the similarity degree of the score for a group of entities; $1 - \eta$ control node $e_i$ and recommended entity $k$ share the proportion of the entity, the proportion is calculated by using the generalized *Jaccard* coefficient Binary Weighting Scheme [35].

The recommended trust generated by the second type is as follows.

$$RT_{ij}^{(2)} = \frac{\sum\limits_{k \in RS_2} Cr_{ik}^{(2)} T_{kj}}{\sum\limits_{k \in RS_2} Cr_{ik}^{(2)}} \tag{7}$$

Combining the trust of two recommendation cases, we get the recommendation trust of $e_i$ to the recommendation entity set about $e_j$ as follows;

$$RT_{ij} = \omega_1 RT_{ij}^{(1)} + \omega_2 RT_{ij}^{(2)} \tag{8}$$

The weights $\omega_1$ and $\omega_2$ of trust $RT_{ij}^{(1)}$, $RT_{ij}^{(2)}$ are calculated as follows:

$$\omega_1 = \frac{|RS_1|^{\frac{3}{2}}}{|RS_1|^{\frac{3}{2}} + |RS_2|}, \omega_2 = \frac{|RS_2|}{|RS_1|^{\frac{3}{2}} + |RS_2|} \tag{9}$$

where $|RS_1|$ and $|RS_2|$ are the order of the two types of recommended entity sets $RS_1$ and $RS_2$, respectively.

*4.3. Global Trust*

Through Equations (3) and (8), we get three trust factors, direct trust $DT_{ij}$, recommendation trust $RT_{ij}^{(1)}$ with interaction with recommendation entity, and recommendation trust $RT_{ij}^{(2)}$ without interaction with recommendation entity. To improve the reliability of trust computing, we propose an adaptive aggregation algorithm to obtain the global trust from node $e_i$ to $e_j$.

$$G_{ij} = \lambda DT_{ij} + \gamma RT_{ij} \tag{10}$$

where $\lambda$ and $\gamma$ are the weights of direct trust and recommended trust, respectively, $\lambda, \gamma \in [0,1]$, and $\lambda + \gamma = 1$.

$$\lambda = \frac{n_1^{\frac{3}{2}}}{n_1^{\frac{3}{2}} + n_2}, \gamma = \frac{n_2}{n_1^{\frac{3}{2}} + n_2} \tag{11}$$

The number of direct interaction between node $e_i$ and node $e_j$ is $n_1$, and the recommended number of other entities to node $e_i$ is $n_2$. The index of $n_1$ is greater than $n_2$, indicating that the direct interaction has a greater impact on the overall trust than the third-party recommendation. In traditional trust calculation, the weight is usually set to (0.5,0.5), which lacks certain adaptability in the weight distribution of trust factors. The trust calculated by this formula not only reflects the interactive experience of node $e_i$ itself, but also comprehensively considers the recommendations of other nodes in the network, which fully reflects the advantages of the multi-source trust model.

The total time complexity of the algorithm is not more than $O(t)$, and the total space complexity is not more than $O(3t)$. The maximum number of cycles of this algorithm is $(1 + k_1 + k_2) * t$, where 1 is the number of direct interactions, $k_1$ is the number of direct recommendations of indirect trust, $k_2$ is the number of indirect recommendations of indirect trust, and $t$ is the number of time stamps. Thus, the time complexity is $O((1 + k_1 + k_2) * t) = O(t)$. The storage space is $O(3t)$, where 3 represents a direct trust value and two indirect trust values, $t$ is the number of times stamp. The time and space complexity of the algorithm proposed in this paper is far less than the existing trust mechanism, for example, the time complexity of fuzzy-based trust models is $O(t^3 long_2 t)$. The trust mechanism proposed in this paper is more lightweight and requires less time overhead.

### 4.4. Trust Update

In the network, there is a common problem that some nodes do not cooperate when completing cooperative tasks, and the non-cooperation between nodes seriously affects the overall effect of the network. Therefore, it is very important to design an effective incentive mechanism to promote the cooperation between nodes.

Trust is dynamic; long-term malicious behavior needs to be reflected in the trust algorithm. After each interaction, it is necessary to update the trust of the interactive entity to more objectively reflect the change in the trust of the interactive entity's trust. If the trust degree of the target node evaluation in this interaction is higher than the trust degree of the estimated node, the trust degree of the node should be appropriately increased, otherwise the trust degree of the evaluated node should be reduced.

If the trust degree of the node before evaluation is $T_{ij}$, the trust value of the entity evaluation in this interaction is $S_{ij}$, and the trust value of the node after the interaction is updated [34] as follows.

$$T_{ij} \leftarrow \theta S_{ij} + (1 - \theta) + \delta \tag{12}$$

Among them, $\theta$ is the trust update adjustment factor and $\delta$ is the trust reward component, which is calculated as follows:

$$\theta = \begin{cases} 1 - \sqrt{\dfrac{\frac{S_{ij}}{T_{ij}} - 0.5}{\frac{S_{ij}}{T_{ij}} + 1}}, & \dfrac{S_{ij}}{T_{ij}} \geq 0.5 \\ 1, & \dfrac{S_{ij}}{T_{ij}} < 0.5 \end{cases} \tag{13}$$

$$\delta = \begin{cases} (1 - e^{-\frac{n}{2a}}) T_{ij} (1 - S_{ij})(S_{ij} - T_{ij}), & S_{ij} \geq T_{ij} \\ (1 - e^{-\frac{n}{a}}) S_{ij} (1 - S_{ij})(S_{ij} - T_{ij}), & S_{ij} < T_{ij} \end{cases} \tag{14}$$

where $a \in [2,5]$ is the number adjustment factor, and $n$ is the number of times the evaluation node $e_i$ interacts directly with the evaluated node $e_j$ in the time window.

The reward and punishment mechanism can prevent excessive reward due to the increase of node satisfaction when swing attack occurs, so as to reduce the impact of the swing attack. The more interactions there are in the time window, the greater is the reward and punishment value of trust.

The ability of this model to resist malicious behavior is mainly dependent on:

1. Normalized evaluation information can suppress the negative effects of malicious nodes being too high or too low.
2. Aggregating direct interactive recommendation trust and similarity-based recommendation trust can effectively reduce the proportion of evaluation provided by malicious nodes in the trust calculation process.
3. Recommendation trust based on *Jaccard* similarity measure can resist the collusion attack of malicious nodes.
4. Trust update and reward punishment algorithm can reasonably restrain the influence of swing nodes, and stimulate the interaction of positive and benign nodes, so that the anti-attack ability of the model is improved. The algorithm of trust update and incentive punishment reasonably restrains the impact of swing attacks, and stimulates the transactions of active and benign nodes, which makes the model's anti-attack ability improved.

Various infrastructures in edge computing with different performance, different identities and different behaviors are modeled as system independent nodes through virtualization, and connected with reliable network through different bandwidth of wired or wireless, forming a comprehensive interconnected large system. Based on this environment, some nodes are often in a mobile state and have strong dynamics. By analyzing the behavior characteristics of network nodes, an effective network trust mechanism is established to improve the security factor of the service and avoid potential security risks during the transaction. The trust evaluation mechanism based on node interaction behavior can effectively suppress the false and fraud behavior of malicious nodes. In Sybil attack, no matter what identity a malicious node enters the network, when the node has malicious behavior, it will be detected, thereby reducing the access permissions of the malicious node.

## 5. Experimental Evaluation and Analysis

In this section, we describe how to build an experimental scheme in a simulated edge computing environment and report the experimental results.

### 5.1. Experimental Methods and Parameters

To verify and analyze the effectiveness of the proposed trust computing scheme, the NetLogo [36] simulator was used in the experiment. NetLogo is a programmable modeling environment for simulating natural and social phenomena. This software is implemented in JAVA and can run on most mainstream platforms (such as Mac, Windows, Linux, etc). It can be easily simulated or created its own model. NetLogo is particularly suited to complex systems that evolve over time. By giving instructions to hundreds of independent agents, we can explore the relationship between individual behavior at the micro level and environmental patterns at the macro level. These macro models are more shown by the interaction between individuals. For comparison, we added Halfweight model [37] and Distributed Reputation Management (DRM) model [25] to the simulator. The Halfweight model provides a flexible way to represent differentiated trust and build trust with multiple aspects of trust. DRM proposes a distributed reputation management system, which uses servers to perform the local reputation management tasks of vehicles in Vehicular Edge Computing (VEC) to ensure the security and improve the network efficiency in the implementation of VEC.

The advantages of this model are mainly reflected in the following aspects:

1. By introducing the time window trust influence factor, the accuracy and rationality of characterizing the trust degree are obviously improved.

2.　　We rationally analyzed the common recommendation situations, abstracted two types of recommendation trust mechanisms, and gave a trust similarity algorithm based on *Jaccard* similarity, which improves the accuracy of model recommendation trust.

To make the experiment closer to the real edge computing environment, two identity-based devices were deployed in the simulator, namely edge devices and edge servers. In the simulation, the entities in the simulator have similar settings [12]. Each entity plays two roles in the system, that is, as a service provider and a feedback evaluator. The node feedback can be of two types: honest nodes (HNs) or malicious nodes (MNs). Honest nodes always provide the right feedback for any entity, and malicious nodes may provide the opposite evaluation of data.

The simulation parameters used in the experiment are given in Table 1. The simulation computer was configured as GPU3.4G, memory 16 GB, and a hard disk 1 TB. The number of edge nodes was 1000, 200 edge devices were in the network. the total time of simulation runs 1000 steps, and the time window of trust calculation was 20 steps.The percentage of malicious nodes (MNs) was set to 20% and 40%. The percentage of collaborative nodes (PCN) was set to 20% and 40%, which means that the edge computing system is relatively idle and busy [38].

**Table 1.** Parameters and their possible value.

| Parameters | Possible Values | Description |
| --- | --- | --- |
| $m$ | 200 | the total number of devices |
| $n$ | 1000 | the number of edge nodes |
| $t$ | 1000 | time-steps of simulation running |
| $\Delta t$ | 20 | time-window for trust computing |
| PCN | 20%, 40% | the percentage of collaborative nodes |
| MN | 20%, 40% | the percentage of malicious nodes |

We designed a centralized performance evaluation mechanism to compare with other trust mechanisms. We mainly evaluated the performance from two aspects: the computational efficiency and reliability of the model as the number of malicious nodes increases.

*5.2. Computational Efficiency Evaluation*

We used global convergence time to evaluate the computational efficiency of the proposed trust mechanism. Global convergence time is the total time of trust aggregation. Global convergence time is helpful to evaluate the computational efficiency of the whole network system. In edge computing, it involves edge servers, a large number of terminals participating in resource access, providing resources to complete computing and data transmission, etc. Because these resources are dynamic and heterogeneous, we regard the convergence time as an important index [39] to measure the stability of the system.

In the experimental node, we believe that it is a relatively stable network environment, that is, most of the service providers can provide good services, and most of the service recommenders can provide honest feedback information. This assumption is also a more realistic network environment. According to this assumption, the proportion of various node types in this group of experiments was set as follows: the percentage of malicious nodes was set to 20% and 40%, indicating that the network network environment is relatively honest and relatively malicious. The percentage of cooperative nodes was set to 20% and 40%. This means that the network environment is relatively idle and busy, respectively. When network nodes are relatively busy, the fast convergence time shows that the algorithm has good computational efficiency.

Figures 4 and 5 show in the comparison of the simulation results of the three models in the node coordination percentage of 20% and 40%. It can be seen that, in this case, the proposed model needs less convergence time than Halfweight model and DRM model, which shows that the proposed model has better computational efficiency and convergence speed. At the same time, it can be seen that

the convergence time of this model increases with the increase in network size, while the slope of Halfweight model and DRM model increases. The results show that, with the increase of network size, the proposed model still has good computational efficiency.

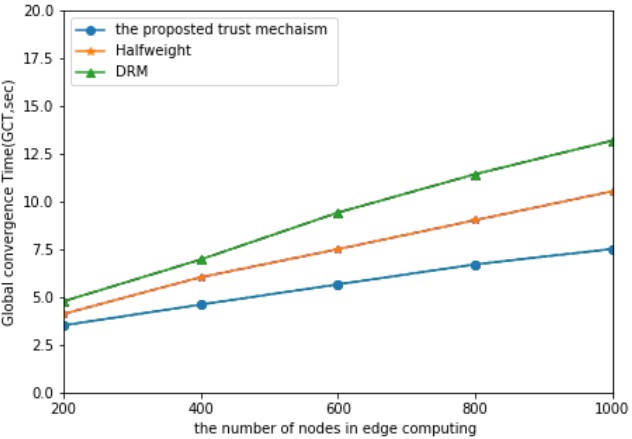

**Figure 4.** The proportion of MNs is 20% and that of PCN is 20%.

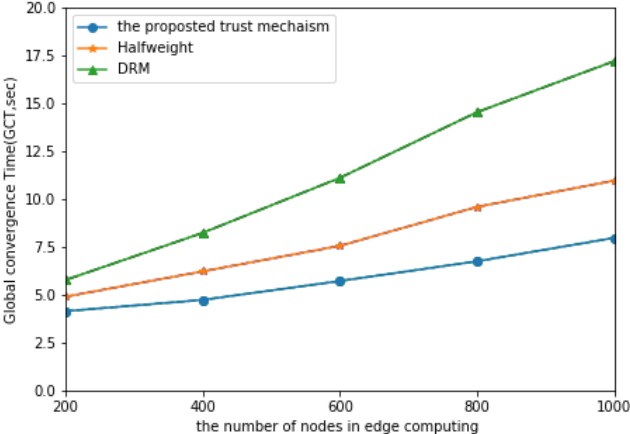

**Figure 5.** The proportion of MNs is 40% and that of PCN is 40%.

### 5.3. Accuracy Assessment

One of the main functions of dynamic trust management is to detect node malicious behavior. Due to the influence of many uncertain factors, errors inevitably exist. The trust mechanism should have strong detection ability of malicious behavior, and the good detection ability reflects that the system has a high Interaction Success Rate (ISR) [40]. Therefore, the accuracy of the model was evaluated according to the ISR. The calculation method is

$$\beta(t_{time-step}) = \frac{1}{t_{time-step}} \sum_{i=1}^{t_{time-step}} H_{ij}(t_{time-step}) \times 100\% \tag{15}$$

where $t_{time-step}$ indicates the total number of runs of the experiment (the total number of time steps), and $H_{ij}(t_{time-step})$ indicates that the $time-step$ determines whether it is successful, returning 1 if successful and 0 if failure. The calculated value of the ISR $t_{time-step}$ reflects the accuracy of the model. The value of the ISR gets closer to 100%, indicating that the model has better accuracy.

In the experiment, the types of feedback nodes were 80% honest feedback and 20% malicious feedback. This value basically conforms to the characteristics of the actual network, because most nodes in the actual network are honest nodes, and only a small number of nodes are malicious nodes. That is, only 20% of the nodes often refuse service.

Figure 6 shows the comparison of ISR simulation results of three models with 20% malicious nodes. From the calculation results, we can see that the three models basically show high ISR, with an average of more than 90%. However, in the three models, the ISR of the proposed model is higher than that of the other two models, and the ISR of the DRM model is the lowest. The above calculation results show that, in a network environment with low malicious node ratio, the three models basically show relatively stable and reliable performance of trust evaluation.

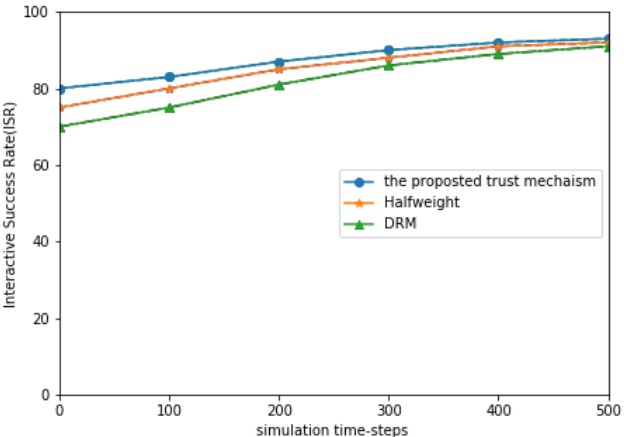

**Figure 6.** The proportion of MNs is 20% and that of PCN is 20%.

The experiment was continued, by increasing the proportion of malicious nodes and observing the ISR indicators of the three models. Figure 7 shows the comparison of ISR simulation results of three models with 40% malicious nodes. From the calculation results, we can see that the ISR of the three models has been reduced, but the rates of reduction of the three models are obviously different. It can be seen that, in a network environment with a high ratio of malicious nodes, the proposed model ISR performance is the most stable, the Halfweight model is second, and the DRM model is the lowest.

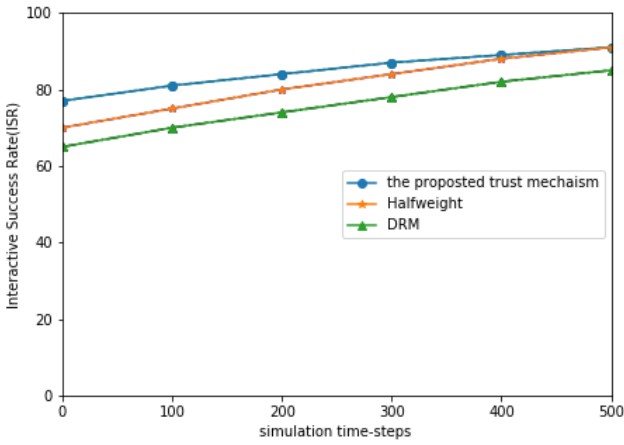

**Figure 7.** The proportion of MNs is 40% and that of PCN is 40%.

## 6. Conclusions

Edge computing will not only be applied to certain vertical industries, but also create a new era infrastructure similar to 5G, which will promote the maturity of more applications. To solve the malicious behavior of nodes, the trust mechanism is introduced to better ensure the security and cooperation of edge computing network environment. This paper proposes a multi-source trust fusion mechanism based on time decay for edge computing. The main content of the trust mechanism includes collecting the credibility of nodes, calculating the credibility of nodes according to the collected evaluation records, and deciding whether to interact based on the trust value of the node. Then, we use the trust evaluation mechanism to evaluate the collaborative work among nodes in the edge computing environment. Simulation experiments show that the algorithm has a certain degree of improvement in computational efficiency and interaction success rate over other existing models, which reduces the situation of malicious node deception.

However, the trust mechanism in edge computing still needs much research. The performance evaluation of the model is not standardized. The functions of existing models have different focuses based on different application environments, and the diversity of the methods and means to build the trust model determines the diversity of the trust model. Therefore, it is necessary to standardize the function evaluation parameters of the trust model in order to improve the scalability and universality of the model.

**Author Contributions:** W.K. proposed this idea and participated in the writing of the manuscript; X.L. and L.H. participated in the discussion of technical issues and revised the manuscript; and Y.L. provided help for the revision of this article. All authors have read and agreed to the published version of the manuscript.

**Funding:** This work was supported by the NSFC-General Technology Fundamental Research Joint Fund U1836215, the National Natural Science Foundation of China under Grant 61672111, and Capital Science and Technology Leading Talent Training Project, China (Z191100006119030).

**Conflicts of Interest:** The authors declare that there are no conflict of interests.

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
