# Peer review of "An Efficient and Credible Multi-Source Trust Fusion Mechanism Based on Time Decay for Edge Computing"

_electronics, doi:10.3390/electronics9030502_

Round 1

Reviewer 1 Report

This paper deals with the issue of trust in edge networks. The essence of the paper is that nodes respond to transactions in either a collaborative or malicious manner. If collaborative they respond correctly. If malicious they may respond incorrectly. Over time nodes learn whether or not to trust a node. They introduce a "Jackard Similarity" based method to determine global trust level which fuses together three other trust measures.

The authors attempt to link this to 5G where edge computing will be necessary to meet the URLLC use case but it could be of relevance in other edge computing environments such as Industy 4.0.

The paper is quite theoretical and specialised.

Overall I think the paper has some value. The idea of quantifying trust in such a rigorous manner and applying it to network nodes is quite interesting and (I think) quite novel.

However, there are a number of issues that I would like to see addressed before the paper is accepted:
. The authors "fuse" three types of trust into one metric called "Global Trust". The fusing is via weightings. The weightings are calculated based on distance of interaction. Where did the formular come for determining these weightings? Is it something invented by the authors? If so it needs more justification than is given.

. The explanation of the evaluation is not at all clear. I am left with so many questions after reading this section. In their experimental evaluation they say that Malicious Nodes may provide incorrect answers. What percentage of the time do they do this? How incorrect are the answers? What sorts of questions are asked - simple binary yes/no or ones where there is a continuous range of values? How does the requesting node learn that the answer is incorrect? How long does it take them to learn this?

. The authors compare their method with DRM and Halfweight. Neither of these are explained in the paper as to what they are. There should be a brief explanation (one or two sentences) as to what these are. In the case of DRM the full wording should be used before the acronym ie "Digital Rights Management (DRM)". There is no discussion as to how the DRM or Halfweight are evaluated.

. The linkage to 5G is quite weak. 5G is used as a use case to illustrate the use of the method. I do not think it is necessary to emphasise 5G quite as much as they do. If they do decide to emphasise 5G they need to say a bit more about where in 5G trust is important. At the moment there are three sentences (on the first half of page 2) on 5G and edge computing. If they continue to emphasise 5G they need to talk more about the URLLC use case and the operation of the Cloud RAN and how edge computing will help achieve them.

. The English needs a lot of work. As well as typographical errors ("it's" instead of "its" for example) there is quite a lot of idiosyncratic expression. The paper needs to be proofread by someone with a very good grasp of English. There are many strange unexplained expressions: "Wisdom Alliance of All Things", "In 5G era, all things intelligence association will truly become a reality", "video surveillance and analysis Wait", "witch attacks and bleach attacks" etc.

. Referencing in the paper is quite poor. As well as examples listed previously, "Jackard similarity" seems to play a significant role in the paper but is not explained, nor is there a reference to it.

. The first paragraph of the Conclusion introduces three new use cases which should have been in the Introduction as motivating examples.

Author Response

Dear reviewer,

Thanks very much for your valuable and constructive comments that helped us greatly improve the manuscript. Based on your suggestions, we have extensively revised the paper. Please see the attachment.

Yours sincerely,

Wenping Kong, Xiaoyong Li, Liyang Hou, Yanrong Li

Reviewer 2 Report

Some language improvements are needed, for instance in the abstract, "it's error information" is not correct and a sentence is left unfinished. There are also many issues with blanks and spaces.

The introduction could be shortened, Although it is correct and would be perfect for a PhD dissertation or a book chapter, almost half of the paper consists in a litterature review and introduction.

The results section on the other hand could be extended and made more complete. The figures could also be improved

Author Response

(The authors gave the same response as above.)

Reviewer 3 Report

In this paper, a trust fusion based mechanism is proposed. The
contribution in this paper is said to be the following three points.

1. The proposed mechanism is lightweight.
2. Trust mechanism is based on time decay degree.
3. The proposed mechanism is a multi-source trust mechanism.

However, the proposal in this paper is very close to what is
proposed in the previous work [12] by the same author group.
In [12], point 1 and 3 are already covered, and thus the true
contribution seems to be point 2 only.

The authors should introduce what is proposed in [12], and clarify
the difference between the proposed method and what is proposed
in [12]. The authors should remove what is written in [12] from
contribution section.

In [12], bad-mouthing attacks and collusive cheating seems to be
considered. However, there is no description on what kind of attacks
the proposed method is supposed to be resistant against is not
described.

The paper lacks intuitive explanation on how the proposed method
works. The authors should add intuitive explanation on how the
method protects the network from various threats.

The proposed method seems to be vulnerable against Sybil attacks.
The authors should explain how the method protects the network
when a single device pretends to be many devices, and builds
trust among those virtual nodes.

The authors should explain why they evaluate global
convergence time. It appears that the authors are assuming that
a malicious node is always malicious and repeats cheating.

I don't understand why the proposed method takes subjective
evaluation as direct trust. If it is subjective, there is no such
thing as a bad-mouth attack. An evaluation by a node is always
correct. Why isn't an objective evaluation, such as success rate,
used for this purpose? Please add explanation.

There are many places with strange capitalization and spacing.
The overall English quality is okay.

Author Response

(The authors gave the same response as above.)

Round 2

Reviewer 1 Report

This is a much improved paper. Many of my concerns have been met. However there are still some areas of concern. In particular some referencing still needs to be done as well as some justification for experiment design.

The English, although improved, still needs further work.

Here are some of the points I've identified but there may well be more:

28 "the intelligence connection of things really come true" to "the intelligent connection of things really can come true"

69 "we think" to "we assert" or "we propose that"

102 "2. relevant work" capitals? "2. Relevant Work"?

288 "e_i can get" to "can be obtained from"

387 Still should have a reference to Jaccard similarity.

387 "Jaccard Jackard similarity"

412 "is mainly reflected in. " to "is mainly dependent on:"

447 "DRM (Distributed Reputation Management)". The convention is usually the abbreviation follows the full term ie.  "Distributed Reputation Management (DRM)"

507 You need to justify the 80% / 20% for cooperative / malicious nodes. Do you have a reference? If not then why did you choose this breakdown?

524 "but will become a new era" to "but will create a new era"

Author Response

(The authors gave the same response as above.)

Reviewer 3 Report

All comments are well addressed.

In Definition 8, "Jaccard Jaccard similarity" seems to be a typo.

Author Response

Dear reviewer,

Thanks very much for your valuable and constructive comments that helped us greatly improve the manuscript. Based on your suggestions, we have extensively revised the paper. 

In Definition 8, "Jaccard Jaccard similarity" seems to be a typo

Response: Thanks very much for this suggestion. We have modified it according to your suggestion. (line 357, page 10)

Yours sincerely,

Wenping Kong, Xiaoyong Li, Liyang Hou, Yanrong Li
